# OpenReview forum: "Bayesian scaling laws for in-context learning"
_ICLR.cc/2025/Conference — Submitted to ICLR 2025_

### Official Review · Reviewer_GBRU · 2024-10-31

**Soundness:** 1
**Presentation:** 3
**Contribution:** 2
**Rating:** 5
**Confidence:** 3

**Summary:**

This paper tries to establish several Bayesian-type scaling laws for In-context learning (ICL).

**Strengths:**

It is reasonable to interpret context relationships by Bayesian inference. Besides, whether a scaling law holds is crucial to the extendibility of a large model (LM). Hence, it is a good motivation to establish Bayesian-type scaling laws for ICL.

**Weaknesses:**

1.The biggest problem of this paper is the lack of theoretical support, deduction, illustration, and explanation of the scaling law Eq. 3 (Final scaling law). The Bayesian formula Eq. 2 is a normal one and the deduction of Eq. 2 (Appendix A) is a common technique in Bayesian inference. However, this paper lacks evidence in deducing Eq. 3 by Eq. 2, though Eq. 3 is somewhat heuristic. Eq. 3 seems to appear suddenly without sufficient explanations.

2.As for particular implementation, the authors take logarithms for non-Bayesian cases and Bayesian scaling laws in different places, respectively. For example, the logarithm is taken outside the whole likelihood, which leads to a negative log-likelihood in the Bayesian scaling laws. However, the logarithm is taken inside the likelihood, and a 1 is added in the denominator, in the non-Bayesian cases. Moreover, the torch.clamp is also used to constrain the log-space parameters to the range (−20, 20), which may significantly affect the experimental results.

3.This work lacks analysis of how the error be scaled. Given the scaling law Eq. 3, how would the error be scaled and amplified by the current scheme? It can be seen that the errors of the parameters could be amplified along with K (exponentially), n (exponentially), and M. Even if a logarithm is taken, the error may still be of considerable magnitude. Is there any treatment to suppress or control the error along with such a scaling law?

Considering the above theoretical and experimental reasons, it is difficult to confirm that the proposed Bayesian-type scaling laws really hold in ICL.


**Minor comments**:

There are some incorrect English expressions that should be fixed. For example, “This may be because the compute allocated to...” in Line 497, Page 10.

**Questions:**

Please address the problems raised in the weaknesses part.

---

> ### Author Response · Authors · 2024-11-17
> **Author response**
>
> Thanks for your review! We try to address your questions below:
>
> 1.  Thank you for this feedback on the textual exposition. Eq. (3) adds the ICL efficiency coefficient ($K$) and weight-tying to reduce latent parameter count, as explained in section 3.2. We agree that our explanation of why we did this could be better. We will update the text to explain both, based on the explanations below:
>     * Given some ICL curves for several tasks on a single model, we can easily deduce the diagonal values in $P$ ($P_{x, x}$) since those are the values the ICL curve for each task asymptotically converges to. However, the non-diagonal values are latent and unobserved. Having significantly more latent variables than observed ones for such a small model makes the model liable to overfitting (since there is nothing grounding the latent vars), so we come up with weight-tying schemes to reduce the number of latent vars to fit. The weight tying also makes the comparison with power laws and other baselines fairer since the number of parameters is much reduced.
>     * The efficiency coefficient is explained in section 3.2 to allow for the possibility of multiple and/or noisy updates. Figure 3b shows the empirical utility of K --- it neatly captures the informativeness of the ICL examples in the GINC setting, with longer examples giving higher K. Another interesting experiment could be to compare K on the same task between a base and instruct model.
>
> 2.  Our Appendix C.1 might not be entirely clear here; for all laws, we take the negative log of the whole expression, so we are computing negative-log likelihood in all cases. For the non-Bayesian laws, we can use logarithm rules to simplify the expression and make it more numerically stable, which is why we have ln n in those expressions; we aren't taking the log any differently. To make this clearer, we can provide snippets from our code in that section for the rebuttal version, and we will of course release our code with the final version of the paper.
>     * We empirically tested the clamping and found no significant difference when clamping in (-20, 20) vs. (-10, 10), and picked the looser range. For our tasks, it seems the optimal values are well within the clamped range.
>     * Also note that none of the clamping and log-space transformation is necessary when using Adam as an optimiser, but Adam performs worse across the board than L-BFGS; L-BFGS is just more sensitive to numerical instability.
>
> 3.  $K$ is a learned parameter and so not relevant for this analysis. We do not expect error to blow up with $n$ because the Bayesian laws always converge to some value as $n \to \infty$ and the output probability is constrained to [0, 1]. As for $M$, our weight-tying scheme, which reduces the number of latent variables from O(M^2) to O(M), significantly combats error by reducing the possibility of overfitting. We can mention some of these points in the text but we believe we have already addressed these, and cannot find analogous analyses in prior ICL scaling law work such as Anil et al. (2024).

---

> > ### Comment · Reviewer_GBRU · 2024-11-28
> >
> > I have read the revised version, the authors’ feedback, as well as all the discussions till now. The authors have explained several issues, such as the usages of torch.clamp and negative log-likelihood. However, the key issues regarding the scaling law itself could not dispel my concerns on whether the proposed Bayesian-type scaling law really holds in ICL, or whether this Bayesian-type scaling law is a significant contribution. To my knowledge, the deduction of Theorem 1 is obvious: the left side of Eq.2 is a $p(\sigma|D)$, thus there is one more $p(\sigma|T_m)$ term in the numerator than the denominator (through some basic deduction tricks regarding the Bayesian formula and the law of total probability), especially when the authors assume a very strong (technically) condition that $D\in \Sigma^n$ contains i.i.d samples.
> >
> > To my knowledge, the essence of a scaling law is that when the parameter number of a model increases, the model can approximate any given function in the same order of accuracy. However, Theorem 1, Eqs. 2 and 3, as well as their deductions and illustrations could not evidently verify this point. That is why I raised the question regarding the amplification of errors last time. The authors claim that the Bayesian laws always converge when $n\rightarrow \infty$, but actually it is the mean estimator of the parameters estimated by $n$ times might converge (under very strict statistical assumptions). This can also be seen by the left side of Eq.2, which is an expectation (i.e., the mean estimator, or the mean of $n$ estimations). But after all, the right side of Eq.2 is only one estimation for the $n$-th sampling. I could not see any mechanism involving all the $n$ samples. In general, only when the errors are not amplified in an order of larger than $n$ along with the increase of parameter number, can these errors be suppressed by the mean estimator with $n$ samples. As for the shrinkage of latent variables from $O(M^2)$ to $O(M)$, it does not necessarily help with the scaling law, or else the authors should provide theoretical or practical evidence.
> >
> > Anyway, the experimental results do indicate that the performance of various models has been improved along with the increase of parameter number, and that the Bayesian based negative log-likelihood scheme is equal to or slightly better than other competitors. Hence I would just keep my score at this time.

---

### Official Review · Reviewer_UJig · 2024-11-02

**Soundness:** 3
**Presentation:** 4
**Contribution:** 2
**Rating:** 6
**Confidence:** 4

**Summary:**

This paper sets up “Bayesian scaling laws” which are a form of scaling law derived from the assumption that in-context learning is performed as Bayesian posterior calculation. The paper is building on recent works understanding ICL with a Bayesian framework, especially Xie et al, 2022. The paper first sets up the Bayesian posterior given by a prior and likelihood over the different tasks the model will be trained on. Parameter tying and efficiency coefficients are introduced to build up to the final scaling law. First, this scaling law is fit to trained transformers on GINC, the authors claim superiority of this scaling law over 3 baselines. Next, the paper explores the effect of SFT, and finds that 1) SFT only modifies the task prior and 2) SFT is more superficial with model scaling. DPO is explored similarly, and the authors find that DPO breaks the Bayesian ICL behavior. Finally, 6 ICL tasks are tested on real language models and the scaling law fits are analyzed.

**Strengths:**

Overall, the paper is well motivated, well written and clear. The experiments are all asking very important questions. Here is a list of strengths:

Contributions:

C1: The result clearly separating SFT’s effect on the prior vs in each distribution is interesting and useful to understand how ICL abilities are affected by SFT.

C2: SFT being increasingly superficial (prior change only) on larger models is interesting, and suggests an important phenomena to study as we scale models.

C3: In general, developing interpretable scaling law from first principles is a good and useful idea.

C4: Showing that DPO destroys Bayesian ICL behavior unlike SFT clearly demonstrates a qualitative change depending on the choices at the fine-tuning stage.


Presentation:

P1: The paper is very well written, the motivation of the work, the development of the theory and the experiments are very clearly described.

P2: The paper’s narrative makes important points about why this work is needed, e.g. “Interpretable scaling laws”

**Weaknesses:**

Even though the paper is great at a conceptual level, unfortunately, many claims are weakly justified. I think the paper has enough qualitative contributions and does not require a justification of competence by scaling law “benchmarking”. In fact, I don’t think the paper’s contribution is degraded at all even if the fits went worse, as the importance of this paper, for me, is that the authors developed a framework to fit a scaling law and interpret its terms, allowing qualitative insights of how Bayesian ICL is, not that NRMSE got 0.02 better. I truly believe this paper has great potential, but here are the main weaknesses:

W1: Even though Table 1 is the basis to the claim “our scaling laws exceed or match existing scaling laws”, it isn’t clear how robust this claim is. The best method alternates between the different Bayesian methods, and the gap with the best base method is very thin. The weight tying scheme introduced gives too many degrees of freedom to set up a Bayesian method (I’m not saying too many model parameters, but many ways to set up different methods, e.g. low rank P matrix) and it is unclear how much this choice will affect the results. Also about significance: Will the DPO fit result stay insignificant even if the number of sample is 10x?


W2: The scaling curves and the fits are not shown as plots for the GINC experiments, preventing the reader from assessing the goodness of fit by eye. Adding these plots will also contribute to making the better performance claim more robust.


W3: I do not understand how Figure 5 supports “DPO can use very little training data to successfully suppress the disfavoured HMMs beyond the ability of ICL to recover.” Is the critical part that the disfavoured classes fall to 0.0 and not something close to 0.1 ? If so, I don’t get the “very little training data” part. If it turns out that this is my misunderstanding, I will reflect this to the final evaluation.


W4: I am not sure if I agree that these results support that LLMs behave in a Bayesian way, from the goodness of fit I see in Figure 6, I would rather argue that they behave in a very non-Bayesian way, especially the instruct models. Even if it outperformed the base scaling laws for these experiments, I do not see why this supports that LLMs do Bayesian ICL since the goodness of fit doesn’t seem persuasive.

W5: The parameter tying and introduction of the efficiency coefficient K is not well justified. Could the intuition behind the tying scheme be clarified?

W6: The paper makes multiple claims about model scaling. However, are these models trained for the same number of steps or (theoretical) FLOPs? Given pre-training scaling laws and the general observation that bigger models train faster (when measured in gradient steps), is it possible that bigger models are just effectively better trained? I see this as a major concern, as real LLMs are trained near the compute optimal frontier, and if small models here are undertrained, the results might be confounding model scale and optimization. At the very least, I believe the equal number of FLOPs should be given for the small models instead of equal gradient steps.

Most weaknesses seems to be addressable, I am willing to modify my score if most of these are addressed, since as mentioned above, the paper makes great points.

**Questions:**

Questions:

Q1: Line 121/122: should the prior be $\rho$ instead of $p$? In general I see both $p$ and $\rho$ for prior throughout the paper. Consistency would help, if possible.

Q2: Why is the prior not included in $\mathcal{M}$ in Theorem 1?

Q3: In section 3.1 are the words “task” and “distribution” used interchangeably? Here also consistency would help, if possible.

Q4: Why is NRMSE a better choice than KL divergence from the ground truth probabilities? What would you expect if the calculations were done with KL divergence?

Q5: I guess Figure 2b is for interpolation? It would be better to write this down explicitly.

Q6: (Optional) Would you be able to adjust figure 1’s colors so that it's colorblind friendly?

Q7: It would be good to have a plot showing the scaling law fit itself. (See W2)

---

> ### Author Response · Authors · 2024-11-17
> **Author response**
>
> We really appreciate your comprehensive review! It raises a lot of important questions that we will try our very best to address below. First though, we do want to discuss the framing of our results. We absolutely agree that a difference in NRMSE of 0.02 is negligible (even if statistically significant) and we are not in the business of hill-climbing on error rates anyways.
>
> The point of this work is to take inspiration from all the papers claiming ICL is Bayesian in transformers using purely theoretical proofs, and see whether it holds up in real life and is practically useful. At least on paper, a Bayesian view of post-training explains failure modes like many-shot jailbreaking (if only task priors are adjusted and not task knowledge). The quantitative results are to show that Bayesian approach is competitive and to simply justify its use as an analysis tool for ICL behaviour, and are **not the end goal of the paper**. We agree that the qualitative findings of how DPO/SFT work are the actual phenomenon we want to understand. We can be clearer about this in the introduction and discussion/conclusion.
>
> Now, to address the weaknesses:
>
> 1.  We are happy to relax the claim in the abstract to "our scaling laws match existing scaling laws" since the margin is so negligible.
>     * In table 1, we felt that every variant was justified to include: Bayesian (original) is the exact functional form we derived, while the other two use different weight-tying schemes which seem equally justified given that the original form overparameterised and has too many latent variables to fit (off-diagonals in $P$). Figure 2b additionally shows that all methods are almost the same (and very correlated) on NRMSE, and is additional evidence for our claim. In our discussion (section 6), we are very careful to note that we haven't *proven* Bayesian methods are better. We can be clearer about all of this in our intro; we just want to point out we are already trying to be very careful not to overclaim.
>     * The insignificance in the DPO comparison likely comes from the very very large NRMSE numbers (>1e6 in some cases!) when the DPO training set size is large and the ICL curve collapses. We don't think the insignificance will change given how noisy the fits are in that setting; figure 5a shows how severe the ICL curve collapse is for all the methods.
>
> 2.  This is a great point. We have many examples of such curves and can include them all in the appendix for the various model depths and SFT/DPO settings we tested.
>
> 3.  Yes, what we want to highlight is that DPO *suppressed the negative subdistribution to 0* unlike SFT (such that ICL no longer works on that distribution in some cases), and that DPO training data requirements are *less sensitive to model size* than SFT. We can reword this sentence to be more clear -- we don't mean it uses less training data in general, just that it needs roughly the same amount for both an 8-layer and 3-layer model in our experiments.
>
> 4.  Did you mean that the *base* model ICL curves look unusual? The instruct models seems to have Bayesian-seeming curves here (we can include the learned fit from one of the laws in this plot). We believe the unusual ICL curve for the base model stems from the base models not being trained on the instruct chat template we used, which has some special tokens that were probably not seen in base-model training ([INST] [/INST] etc.). We can report additional ICL curves for Llama 8B with a simpler input template format (e.g. "Question: ... Answer: ...") in an appendix, which we believe should show a more monotonic ICL curve.
>     * Also note that figure 6b is the *posterior* of the Bayesian scoring-wise fit, it's not the actual computed ICL curve. The task posterior seems to be the right way to compare different models since it takes out the noisiness of the conditional probabilities for the task, which is why we show it here. We can plot the actual ICL curve on figure 6a as well to help see this.
>     * In general, we believe that our findings provide additional *evidence* for the claim that LLMs do ICL in a Bayesian way. We are one of the few works that empirically try to test this (as opposed to just theoretically prove it) and we believe this kind of an experiment is necessary in the literature to really assess the claim. If someone finds a non-Bayesian functional form that performs significantly better than ours, that would be empirical evidence against the claim, but so far none of our baselines succeed in this (including the logistic baseline, which is newly introduced by us).
>
> **[continued in the next comment]**

---

> ### Author Response · Authors · 2024-11-17
> **Author response (part 2)**
>
> **[continued from previous comment]**
>
> 5.  We can be clearer about this in section 3.2. Essentially, given some ICL curves for several tasks on a single model, we can easily deduce the diagonal values in $P$ ($P_{x, x}$) since those are the values the ICL curve for each task asymptotically converges to. The non-diagonal values are latent and unobserved. Having significantly more latent variables than observed ones for such a small model makes the model liable to overfitting (since there is nothing grounding the latent vars), so we come up with weight-tying schemes to reduce the number of latent vars to fit. The weight tying also makes the comparison with power laws and other baselines fairer since the number of parameters is much reduced.
>     * The efficiency coefficient is explained in section 3.2 to allow for the possibility of multiple and/or noisy updates. Figure 3b shows the empirical utility of K --- it neatly captures the informativeness of the ICL examples in the GINC setting, with longer examples giving higher K. Another interesting experiment could be to compare K on the same task between a base and instruct model.
>
> 6.  This is a very good point. For the current experiments, the models are gradient-step matched, not FLOPs-matched. They trained to convergence given the data amount we used (we can add loss curves to the appendix to show this). The FLOPs-matched setting does seem interesting though; since the models are quite small, we are happy to run some FLOPs-matched experiments and see if the loss and ICL behaviour is substantially different; if they are, we could replicate our SFT/DPO experiments on those. We would keep these in an appendix.
>     * For SFT/DPO we believe gradient-step matching is more realistic since the bottleneck in these settings is usually data amount, not compute; a practitioner could ask, given some fixed dataset, what scale model should I finetune to avoid many-shot jailbreaking?
>
> As for the additional questions:
>
> 1.  You are correct, that should be $\rho$. We will carefully go over the math in the text again to check our notation.
>
> 2.  This is a typo, as above (it should be included and we will fix this).
>
> 3.  Yes. We will standardise this to "task" to be clear.
>
> 4.  For ICL, the practically useful variable is the probability of the correct answer. Additionally, the Bayesian derivation directly operates on probabilities, and previous work (such as Anil. et al, 2024) also came up with functional forms to fit to probability or log probability. We did consider KL divergence, but unlike e.g. for the language-modelling objective, it isn't the main metric of interest when we are studying ICL. KL divergence is also unbounded and thus less numerically stable to fit than the probability (which is in [0, 1]). If we simply modify our functional form and the baselines to express KL divergence from ground truth, we don't expect to see any difference, but keeping the fits stable could be much harder.
>
> 5.  Yes, thanks for catching this, we will specify that.
>
> 6.  Oh this is a very good point that we had not considered. We can add texture to the boxes to make this clearer for color-blind readers.
>
> 7.  We agree! We can supply many of these for the GINC experiments in the Appendix, and add them to figure 6a as well.
>
> Thanks again for this very insightful and constructive review!

---

> ### Comment · Reviewer_UJig · 2024-11-23
>
> Thank you all for the reply! In general, the updated manuscript will help a lot!
>
> 1. The updated manuscript will help, thank you!
>
> 2. Same as 1.
>
> 3. I see, this explanation makes sense. I think it would be better if the following is highlighted in the updated manuscript: "DPO suppresses the negative subdistribution to 0" and some quantification of how close to zero it is will help! Irrespective of the author's statement that "DPO training data requirements are less sensitive to model size than SFT." However, does the claim "Unlike SFT, DPO can use very little training data to successfully suppress the disfavoured HMMs beyond the ability of ICL to recover" in l. 418,419, stand?
>
> 4.1 Yes the manuscript update showing the goodness of fit will help me understand this better, currently its hard to see how good the fit is. I am also a bit confused why the authors decided to (or had to) use special tokens to evaluate the base model.
>
> 4.2 I totally get the argument: "We are one of the few works that empirically try to test this (as opposed to just theoretically prove it) and we believe this kind of an experiment is necessary in the literature to really assess the claim." and agree that this empirical validation is exactly what we need beyond theoretical possibility claims. This by itself is an important contribution of the paper. My main concern is whether Table 2. actually supports "our findings provide additional evidence for the claim that LLMs do ICL in a Bayesian way.", as it seems like the logistic baseline does at least as good? Shouldn't the claim be "our findings suggest that Bayesian assumptions are as consistent as conventional scaling laws for ICL in LLMs do ICL"? Please tell me if I am misunderstanding something!
>
> 5. Sorry that the question was vague. I'm wondering what justifies this specific scheme of weight tying. At the same time, I can understand why this is the most reasonable thing to do. If you can motivate this better it would be great, but I don't think this is a weakness!
>
> 5.1 I'm a bit afraid if K will absorb any non-Bayesianness and thus make this scaling law fit a non-Bayesian ICL process too well.
>
> 6. Yes! Even showing a single FLOPs matched run (without re-running everything) would be great. But also, perhaps you can just take an earlier checkpoint of your bigger model so that you don't need to re-run? But if the authors are pretty sure the loss is saturated, I think this is uneeded.
>
> 6.1 Yes, totally, the question was only about the pre-trained models.
>
> ------
> **Summary**
> In general, I think I am willing to raise my score if I can see *some* of these: (more important on top)
> - The updated manuscript addressing 1,2,3,4
> - Clarification of why Table 2 supports: "our findings provide additional evidence for the claim that LLMs do ICL in a Bayesian way."
> - Explanation on the claim: "Unlike SFT, DPO can use very little training data to successfully suppress the disfavoured HMMs beyond the ability of ICL to recover"
> - A single FLOPs matched experiment (if the loss is not saturated)
> - Does the presence of the K parameter affect if we can draw conclusions about Bayesianness from these scaling laws?
>
> Thank you!!!

---

> > ### Author Response · Authors · 2024-11-28
> >
> > Thank you so much for your response, we really appreciate how responsive you've been! Since the rebuttal revision deadline is soon, we have done our best to update the manuscript (given travel during Thanksgiving break etc.) to address your comments; we will continue to engage over OpenReview after that deadline passes.
> >
> > Responding point-by-point again:
> > 1. Great, we have updated the abstract and various discussion/conclusion sections in the paper to adjust our claim.
> > 2. Page 27 onwards have detailed figures of the actual ICL curves for GINC models. Let us know if you want more figures, we are happy to provide them in OpenReview after the rebuttal revision deadline passes. The DPO ICL curves are particularly interesting since they clearly show collapse of the ICL curve, which ends up very non-monotonic in most cases.
> > 3. We have reworded this part to be clearer. The "very little training data" part of the claim is not true relative to SFT; what is true is that DPO suppresses ICL of disfavoured distributions much more strongly than SFT.
> > 4. Responses:
> >     1. We updated this plot to show the Bayesian (scoring-wise) fit. We used the special tokens for consistency since it was arbitrary to pick some other format for the base model (and we had to use the instruct format for the instruct model). The consistency at least keeps the comparison fair. If we get time to run experiments with other templates we will certainly let you know.
> >     2. We agree on this. We relaxed the claim "our findings provide additional evidence for the claim that LLMs do ICL in a Bayesian way" in the discussion section. At best, we have shown that a Bayesian interpretation of ICL is not incompatible with reality; we certainly do not want to claim that we have *proven* that ICL is Bayesian (see "Are LLMs Bayesian?" in the discussion) so rewording this seemed right. I think we all agree on this point, it was just the wording was perhaps a bit strong in the original version.
> > 5. We added some explanation in section 3.2 to clear this up, the main reason to tie non-diagonal parameters is that they are latent. Having less latent parameters to fit reduces the risk of overfitting.
> >     1. $K$ merely squishes/stretches the fit along the x-axis, it does not change the *shape* of the curve. E.g. $K=2$ means that updates are done twice as fast. This doesn't change that the update rule is Bayesian. We think we have some reasonable justification for why we did this, and the correspondence of $K$ to the example length in the GINC setting really supports the claim that $K$ empirically captures the informativeness of the ICL examples.
> > 6. Here is the loss curve from some earlier logs (missing 3 and 16-layers; image is anonymised): https://i.imgur.com/WkCspGv.png. The loss seems saturated for >=4 layers in this plot, so we don't expect FLOPs-matched experiments to be different in this regard; it really does seem that larger models are slightly better than smaller ones. We do have logs for checkpoints of this models; we can reply with some additional results in a few days if we find something interesting on this point.
> >
> > Thanks so much for your thoughtful comments again! The new version of the manuscript is up.

---

> ### Comment · Reviewer_UJig · 2024-11-29
> **About W6 (1/2)**
>
> I think most concerns(1~5) are very well addressed. However, I am unsure how to handle the weakness 6. It would be great if we can have a discussion on this, I am really ready to raise the score as soon as W6 is addressed!
>
> ---
>
> 1. Thank you!
> 2. My main concern was that the scaling law fit wasn't visible like in Fig 6. Quoting my original comment: "The scaling curves and the fits are not shown as plots for the GINC experiments". The DPO's ICL collapse is indeed interesting!
> 3. Thank you this really makes it much easier to understand with respect to the plot.
> 4. Thank you, the discussion is especially very clear.
> 5. Thank you!
> ---
> ---
>
> 6. My interpretation of these loss curves is as follows:
> - **Interpretation 1: FLOPs is indeed a confounding factor:** It seems like it is precisely the case that FLOPs is being a confounding factor. In fact, if you can see that the loss of the 1layer model at epoch 4 is almost precisely the loss of the 4layer model at epoch 1. Same for the 4layer model at epoch 4 and the 8 layer model at epoch 2. Of course, the simple linear relation I'm pointing out wouldn't hold either robustly across scales because of 1) early training dynamics and 2) scaling laws [1,2] themselves don't predict equal flops=equal loss.
> - **Interpretation 2: Going from 1-layer to 4-layer gives a qualitative change:** I see that there is a *qualitative* change between 1,2 layer models and 4~12 layer models, as one can see from the crossover of the 1,2 layer models losses.
> - **Interpretation 3: Saturation in 8~12 layer models:** I agree to the authors that model 8~12 are likely saturated, *maybe* the 4-layer model is as well.
>
> ---
> **[Continued below...]**

---

> ### Comment · Reviewer_UJig · 2024-11-29
> **About W6 (2/2)**
>
> **My concern from these interpretations is as follows:**
>
> In general, I think conclusions about "model scaling" should be drawn in 1) equal flops or compute optimal setting 2) in a regime where scaling is smooth in loss without qualitative changes. This is because without these conditions one can easily confound the effect of scaling the model and just training more. e.g. see Figure 2 in Kirsch et al [3] where the model size is scaled keeping the # of steps equal and there seem to be a qualitative change.
>
> The *main text* in general and Figure 2b, 3b, 4b, 5b seem to suggest the plots show an effect of model scaling in the sense of LLM scaling laws (Kaplan et al, Hoffmann et al), where we have a smooth and predictable scaling curve of loss without qualitative changes depending on architecture. (In fact, Kaplan et al further strengthens this latter argument by showing that "adjusting the width or architectures while holding the # of params. constant don't have a significant effect".) However, interpretation 1 and 2 suggests that in these experiments 1) models are not trained at the compute optimal frontier (or even at the equal FLOPs threshold) 2) there is a qualitative change depending on the number of layers. Thus, drawing conclusions about "model scaling" from Figure 2b, 3b, 4b, 5b seems like its confounding scaling and more training and qualitative changes. The authors suggest that the 4~12 layer models are saturated. If all conclusions are drawn from saturated models without qualitative changes, this will indeed refute the concern. However the scaling findings in Figure 2b, 3b, 4b, 5b, seem to heavily rely on the 1,2,4 layer models. Perhaps the only *pure scaling* result seen here is the change between the 4-layer and 8-layer model.
>
> **I think there are two options (but feel free to suggest something else):**
> - **1. Clarification in the manuscript that these are *equal steps*:** I think this is the cleanest way forward. There is nothing *wrong* about studying model scaling with equal steps, as long as this is *clear*. The problem is that sometimes this causes the audience to interpret results as "bigger models are fundamentally different" even when the difference can be explained by them having received simply more training. Thus I think the paper should clearly recognize that bigger models just train faster per steps, but equal number of steps are given in this work, and clearly show the loss curve.
>
> - **2. Reproducing results with equal loss or equal FLOPs.** I think 1 is much easier, but I'm adding this **mostly to clarify what I think a "model scaling" result should look like.** Models which do not show qualitative changes (4layers to 12layers) should be assembled and checkpoints should be collected at equal FLOPs *or* compute frontier (which will require a pre-training scaling law experiment, and thus is probably out of scope). Then the results should be drawn from these checkpoints. Alternatively, one can study different checkpoints of a single model and show that the effect of scaling compute and model scale are qualitatively different, refuting the hypothesis that "model scaling effects can be simply explained by more compute given to bigger models".
>
> ---
>
> I really like the idea of the paper, and again, I am ready to increase the score as soon as W6 is addressed. The fact that we are even able to have this discussion about "are bigger models more Bayesian?" is only possible because of the framework in this paper. But I really think this issue about model scaling vs simply more FLOPs is given should be clarified.
>
> Please tell me if any argument I made is unclear, I am happy to explain it further!
>
> Furthermore, given that the manuscript cannot be updated, I will trust any result reported from the authors without plot evidence, especially since the authors has already demonstrated their honesty by showing the loss curves very clearly.
>
> ---
> [1] Kaplan et al
> [2] Hoffmann et al
> [3]https://arxiv.org/pdf/2212.04458

---

> > ### Comment · Reviewer_UJig · 2024-11-30
> >
> > I'm just checking in for this, wondering if the authors would be able to either:
> >
> > 1. Clearly mention in the manuscript that the scaling results are equal steps, and bigger model are effectively better optimized.
> >
> > 2. Persuade me why equal step comparison is fine to present the results as model scaling result, or still delivers useful insights about model scaling. (I wrote "persuade" as this is not something one can prove but more opinionated, for me a "model scaling" result means "an effect purely caused by the size of the model", but if the authors believe the equal step setup is still a good setup for "model scaling" conclusions, it would be good to hear why)
> >
> > 3. Show evidence that the results between layer 4-12 still reproduces the claims even when normalizing for flops (needs additional plot making... so seems a bit exhaustive given the time).
> >
> > I will raise my score immediately if the authors can do any of these (or do anything else appropriate to address this concern).
> >
> > Thank you!

---

> > > ### Author Response · Authors · 2024-12-01
> > >
> > > Thanks for the response! You raise reasonable points. We are preparing to run a few FLOPs-matched pretraining experiments on GINC, but first to address some points:
> > > * In general, our use of *scaling* in the title is in reference to the scaling of *inference-time* compute. This is not out-of-place in the literature, see e.g. [Snell et al., 2024]. The ICL curve which we seek to model is a relationship between number of ICL examples *at inference* and expected NLL of the correct answer. Our scaling law (as well as the baselines we compare against) doesn't include any terms relating to model size/depth/etc. Our main claims are about the shape of the ICL curve regardless of model size (and indeed independent of any properties of the model).
> > > * **However**, you are right that in the GINC figures and main text, we make claims regarding how models of different sizes behave under ICL. Having read your responses, we agree that there is a confound here about the larger amount of compute that these bigger models get in order to learn the data distribution. Our logs only contain statistics over training and not stored checkpoints (due to disk space limitations), but we are prepared to run FLOPs-matched experiments to see how the results differ, since the model training is small-scale and not expensive. We will outline this below.
> > > * Still, there are reasons that a training steps-matched set of experiments is compelling. For example, under a Bayesian perspective, this means all the models got the same evidence (=same training data) to arrive at their prior. Additionally, considering real-world study of transformer LMs, the [Pythia](https://proceedings.mlr.press/v202/biderman23a/biderman23a.pdf) model series has shown that steps-matched comparisons can allow for very interesting and enlightening experiments, showing that larger models memorise more, are more sensitive to training corpus modifications (their experiment on de-gendering pronouns in pretraining), etc. Later work has done some nice training dynamics experiments on these models.
> > > * Also note that the paper whose pretraining setup we replicate ([Xie et al., 2022](https://arxiv.org/abs/2111.02080)) uses steps-matched training. It seems that the issue of confounding scale with FLOPs in this setup was not mentioned or considered at the time. This is not to justify inertia on our part, but merely to point out that we started by considering the setup in that paper; now that we know more about scaling laws, we definitely see the justification for changing the setup.
> > > * **We will definitely mention in the manuscript that experiments are equal steps and include some of the compelling reasons for steps-matched experiments as given above, alongside FLOPs-matched experiments.** We basically think that both settings are valuable to study.
> > >
> > > ## FLOPs-matched experimental setup
> > > * Currently, using the library `calflops` and computing single-forward pass FLOPs with inputs of shape `fwd = (batch_sz * grad_acc_steps, 1024)`, total forward-pass FLOPs as `tot_fwd = fwd * num_train_epochs * (len(train_dataset) //  (batch_sz * grad_acc_steps))`, and total forward + backward FLOPs as `3 * tot_fwd` (which is an approximation), for the current experiments we get:
> > > | Model depth | 1 forward pass FLOPs | Total FLOPs (fwd + bwd) |
> > > | :-- | --: | --: |
> > > | 1 | 0.117T | 218.837T |
> > > | 2 | 0.233T | 436.387T |
> > > | 3 | 0.349T | 653.938T |
> > > | 4 | 0.465T | 871.489T |
> > > | 8 | 0.929T | 1741.691T |
> > > | 12 | 1.393T | 2611.894T |
> > > | 16 | 1.857T | 3482.097T |
> > > * This matches up with the rough math in [Anthony et al., 2023](https://blog.eleuther.ai/transformer-math/), showing linear scaling of compute relative to model parameters (when dataset size is fixed).
> > > * What we propose doing then is training each of the models to roughly 3482.097T FLOPs, which seems to saturate the 16-layer model. We will train for 1 epoch (so not repeating data for 5 epochs as in our current setup) and simply scale up the dataset size by a factor of `num_params_16layer / num_params_[n]layer` for each model (so the 1-layer model will train on 16x the data of the 16-layer model, 2 layer on 8x, etc.)
> > > * These results should be ready by tomorrow and we will report pretraining results / ICL law fits to you in a follow-up comment.
> > >
> > > Thanks again for engaging so well with our work; we believe it has been a useful experience for us!
> > >
> > >
> > > [Snell et al., 2024]: https://arxiv.org/abs/2408.03314

---

> > > > ### Comment · Reviewer_UJig · 2024-12-01
> > > >
> > > > Thank you for the response!
> > > >
> > > > > In general, our use of scaling in the title is in reference to the scaling of inference-time compute.
> > > >
> > > > Yes! I was not confusing the inference scaling claims, my comments were indeed focused on specific claims about model size scaling.
> > > >
> > > > > Model Scaling Claims
> > > >
> > > > Oh, wow, I didn't intend to actually make the authors run experiments (I always thought just clarifying in the text would suffice). But then of course, I'm genuinely curious whether the claims will hold for FLOPs normalized runs. (And it seems like the authors are as well.)
> > > >
> > > > > Scaling with fixed steps experiments
> > > >
> > > > I think the "same evidence" can be achieved by multi-epoch training on smaller models (so they see the same evidence while getting the same amount of *optimization*). However, I totally agree that steps fixed experiments can be interesting in general! However even in the pythia example I think it is important to *clarify* that there *is* an optimization effect here.
> > > >
> > > > > Xie et al 2022
> > > >
> > > > I have the exact same criticism for Xie et al 2022 as well: Xie et al Figure 6 shows that the pretraining loss is the same for 12-layer and 16-layer models, but not for the 4-layer model. Interestingly, one can see exactly the same criticism I had for this work's Fig 2b, 3b, 4b, 5b, for Xie et al's Figure 5: The ICL accuracy rises a lot for 4-layer to 12-layer but only slightly for 12-layer to 16-layer.
> > > >
> > > > > Clarification for steps!
> > > >
> > > > Thank you very much! By this alone, I will raise my score!
> > > >
> > > > ---
> > > > **Experimental setup.**
> > > >
> > > > Yes, I think the FLOPS=6*D*N generally works (Hoffman et al [https://arxiv.org/abs/2203.15556], Table A4), and it makes sense that adding layers will linearly increase flops (not counting the embedding/unembedding parameters or flops)
> > > >
> > > > The rest of the setup makes totally sense!
> > > >
> > > >
> > > > ---
> > > > **Summary:** It is clear that the authors are treating this concern seriously and rigorously. I will raise my score, and it would be good to know how the FLOPs normalized experiments turned out! **But not having these experiments (or these experiments fully explaining the scaling results) will not change my assessment, as the main point was that this should be clarified, *not* that this is wrong.**

---

> > > > > ### Author Response · Authors · 2024-12-04
> > > > >
> > > > > Thank you for all the insightful discussion and raising the score! We definitely believe that paper has improved from your engagement. We will update the text as discussed but we also want to tell you the results of the FLOPs-matched experiments since they were quite interesting.
> > > > >
> > > > > ---
> > > > >
> > > > > We ran the pretraining for 4, 8, 12, and 16 layers as outlined in the plan above (so 4-layer model was trained for 4x the data as the 16-layer model). Due to some memory issues on our cluster that we will leave for debugging later, we weren't able to train the 1, 2, and 3-layer models, but these results already are quite informative in our view. We also didn't tune hyperparameters.
> > > > >
> > > > > Based on the loss numbers on train and val reported below, it indeed seems that the 4-layer model was undertrained, while 8, 12, and 16 were saturated. Only the 4-layer is noticeably lower than in figure 2b of the paper. We did originally suspect that Xie et al. undertrained their 4-layer model, but since our loss numbers already looked better than theirs in the steps-matched setting we assumed it was due to suboptimal hyperparamater selection. However, this is evidence that more training data was still needed for the model.
> > > > >
> > > > > | Model       | Train loss | Val loss |
> > > > > |-------------|---------|---------|
> > > > > | `gpt`₄      | 1.326   | 1.325   |
> > > > > | `gpt`₈      | 1.310   | 1.366   |
> > > > > | `gpt`₁₂     | 1.307   | 1.330   |
> > > > > | `gpt`₁₆     | 1.331   | 1.334   |
> > > > >
> > > > > Now we replicate some of the figures from section 4.1 under the FLOPs-matched setup.
> > > > >
> > > > > First, [figure 3b](https://i.imgur.com/ywIDGMg.png) (anonymised), which plots ICL efficiency against model scale, faceted by ICL example length. The observation that longer ICL examples in GINC lead to greater efficiency basically holds; some of the fits are noisy (perhaps we should average over a few fits in the paper). **However**, the 4-layer model, if anything, has better ICL efficiency than the other models! This means our scaling claim in this section is unsupported; it's only true in the steps-matched setting, and only because the 4-layer model was undertrained (and perhaps the smaller models as well).
> > > > >
> > > > > The other relevant plot is [figure 2b](https://i.imgur.com/fnVD537.png) (anonymised), which plots the NRMSE of the various laws and baselines against model scale. Nothing is substantially different in this plot, except that NRMSE for the 4-layer model is much lower; it does seem that all of these scaling laws are worse at modelling undertrained models. The Bayesian laws do hold up generally.
> > > > >
> > > > > We weren't able to run SFT or DPO experiments in this setting yet, but will do so for the final version (and replicate all the plots in this setting). Overall though, we think this set of experiments was/will be very interesting and we appreciate your insistence on qualifying our claims about scaling and giving us the idea to check the FLOPs-matched setting! We will report these experiments in an appendix and change the main text to not make scaling claims without referring to the FLOPs-matched experiments as well. Although it changes some of our conclusions (and those of Xie et al.) about model scale and ICL, we do think this adds evidence for the Bayesian ICL laws being a useful tool for understanding the ICL behaviour of these models.
> > > > >
> > > > > Thank you so much again for the review and continued engagement!

---

### Official Review · Reviewer_Xv3b · 2024-11-03

**Soundness:** 2
**Presentation:** 3
**Contribution:** 3
**Rating:** 5
**Confidence:** 2

**Summary:**

This paper proposes a Bayesian scaling law for in-context learning (ICL). Building on prior work that suggests ICL can be understood through a Bayesian framework, the authors relate the number of in-context examples to the probability of predicting the next example. Essentially, the paper iteratively applies Bayes' theorem to model how ICL updates the task prior as new examples are encountered, ultimately deriving a functional form for the posterior distribution. Experimental results are reported on safety alignment tasks, where many-shot prompts are used to demonstrate the predictive ability of the Bayesian scaling law, including cases where prompts can bypass safeguards. The paper also compares the Bayesian scaling law to a baseline power law model, evaluating performance based on interpolation and extrapolation error, and demonstrates the superior predictive accuracy of the Bayesian approach.

**Strengths:**

Overall, this is a very interesting paper, as the Bayesian scaling law offers insights into phenomena such as many-shot prompt jailbreaking. If the conclusions of this work hold, they could strengthen the argument that ICL operates in a Bayesian manner.

**Weaknesses:**

My primary concern lies with the functional form of the posterior expectation in Equation (2). Specifically, in Equation (2), over which random variables is the expectation taken? The notation used here is somewhat unclear to me. Additionally, in Equation (17) of the appendix, the authors appear to apply the linearity of expectation, which seems to assume that E(A/B)=E(A)/E(B). Am I missing something here?

**Questions:**

See my comments on weaknesses

---

> ### Author Response · Authors · 2024-11-17
> **Author response**
>
> Thank you for the review! We appreciate that you took the time to go through the derivation in the appendix and found your review very helpful.
>
> In (2) we mean to say that the symbols in document $D$ are sampled from $\lambda$; we will change the notation to say $D \sim \lambda$ and clarify this.
>
> We agree that the current version has an error  in (17) since we E[A/B] only = E[A]/E[B] for particular distributions of A and B (cf. https://link.springer.com/article/10.1007/BF02927114), which we have not proven (or claimed) for our A and B. **We have fixed this in the rebuttal response paper; we also  outline here how we did so. Our fix requires no rerunning of experiments and is purely formal.**
>
> In our theorem, we want to model the effect of repeated Bayesian updates when each piece of evidence is sampled from $\lambda$. When fitting the functional form, we set $\lambda$ to one of the subdistributions of interest, but that makes the expectation $\mathbb{E}_{\sigma \sim \lambda}[A/B]$ unsimplifiable (without further assumptions about the distribution of A and B).
>
> Instead of being stuck here, we can assume that $\lambda$ is one-hot, so that each ICL example is identical and thus interchangeable. Since $\lambda$ now collapses to having a single outcome (repeated ICL symbols A,A,A,...), we can simply get rid of the expectation. The idea here is that we have noise in the ICL process modelled entirely by $\delta$ (the probability of an ICL symbol conditional on a task) and not by the document sampled from $\lambda$. Intuitively, this simplification means that we are now finding the *average* Bayesian update needed to get the ICL curve we are trying to fit.
>
> There are some possible next steps that we have left out of this work: one could introduce some assumptions about the values in $\delta$ to somehow make (17) simplifiable without collapsing the expectation to one outcome. We are aware of some unpublished work trying to do this for inference-time scaling laws. We believe that for the purposes of our work, the simplification of $\lambda$ is sufficient, and leave further modification to future work.
>
> We would also like to point out that previous work on ICL curve fitting has not used this kind of theoretical justification to obtain their functional forms (see our discussion in appendix B), and so we are facing a novel problem when trying to derive a functional form from theoretical claims that ICL is Bayesian---we believe that doing so is the only way to really test that claim empirically.

---

> > ### Author Response · Authors · 2024-11-28
> >
> > Commenting to let you know that we have fixed the proof in the updated rebuttal revision of this paper. We hope this version resolves your concerns!
> >
> > For future work, we do leave the question open of adding additional assumptions to the distribution of symbol probabilities to model the ICL curve. As we said before, we are aware of some research trying to do this for inference-time scaling laws, so this may end up being an interesting line of work.

---

### Official Review · Reviewer_pmG2 · 2024-11-04

**Soundness:** 3
**Presentation:** 3
**Contribution:** 3
**Rating:** 8
**Confidence:** 3

**Summary:**

This paper makes progress towards understanding (1) scaling laws for ICL and (2) understanding whether ICL is bayesian. In particular, it presents a scaling law for ICL via Bayesian assumptions, and validates these scaling laws using ICL behavior for small LMs trained on controlled synthetic data as well as LLMs trained on natural language.

**Strengths:**

1. This work investigates an interesting problem. Based on some assumptions, it proposes a new theory and validates this theory with experiments. The work also does a good job comparing the proposed scaling law to prior works.
2. I appreciate that the experiments include both toy settings as well as real world LLMs.

**Weaknesses:**

The paper primarily focuses on transformers. However, it would be interesting to see some experiments for e.g., state space models, and whether ICL is bayesian in these models as well.

**Questions:**

I found the observation that ICL efficiency (K) roughly increases with model depth to be interesting. Do the authors have any insights about the role of model width?

---

> ### Author Response · Authors · 2024-11-17
> **Author response**
>
> Thank you for the positive review! To address your points:
>
> 1. It would be very interesting to see if these results hold for alternative architectures. Previous theoretical work on the learning algorithm underlying ICL, particularly those claiming that it is Bayesian, has largely focused on transformers (see section 2 of our paper) and so we prioritised studying that architecture. Future empirical work on ICL in other architectures would be great once we have some theoretical proofs to start with.
> 2. Yes, in our experiments we did not study the effect of model width; all model hyperparameters were held constant besides depth. We don’t expect width to have much of an effect since (1) the synthetic task we train on is very simple and is learned well by our small models (when depth is at least 3), and (2) ICL has been tied to model depth in previous work via induction heads (e.g. https://transformer-circuits.pub/2022/in-context-learning-and-induction-heads/index.html) which require model depth of at least 2 to develop.

---

> > ### Comment · Reviewer_pmG2 · 2024-12-02
> >
> > Thank you for your response! I will maintain my score.

---

### Meta-Review · Area_Chair_q9xA · 2024-12-21

**Metareview:**

This paper proposes a Bayesian scaling law for in-context learning. The method is motivated by theory, and the paper conducts empirical evaluations to verify the fitting ability of the proposed scaling law. The experiments are conducted by firstly suppressing certain capability of an LLM through SFT and then using ICL to bring this capability back.

The reviewers in general praised the paper in terms of attempting to derive the proposed scaling law from theory, the insights and interpretations offered by the proposed method, and its potential to strengthen the argument that ICL behaves in a Bayesian way.

However, the reviewers also expressed some important concerns about the proposed scaling law. One major concern is whether the proposed Bayesian scaling law is a good fit to the empirical observations, and whether it is better than the non-Bayesian baselines. This is pointed out by both Reviewer UJig and Reviewer GBRU. Reviewer UJig also had extensive discussions with the authors during rebuttal phase regarding whether the scaling law for different architectures should be equal-steps or equal-FLOPs. After the discussions, the concern of whether the proposed scaling law serves as a good fit still remains.

Another concern is the lack of rigorous theoretical support as pointed out by Reviewer GBRU. This concern is also shared by Reviewer Xv3b, who pointed out an error in the original theoretical derivation. The authors proposed a method to resolve this error by imposing an additional assumption. However, in my opinion, this additional assumption may be too restrictive, because it implies that "each ICL example is identical and thus interchangeable". This may also limit the practicality of the theoretical derivations and hence make the proposed scaling law more heuristic in nature.

Given these concerns, rejection is recommended. The paper has taken an interesting approach and we do see the potential of this work, so we encourage the authors to take into account the reviewer comments to further improve the paper.

**Additional Comments On Reviewer Discussion:**

During the rebuttal and discussion, the reviewers (especially Reviewer UJig) had long and fruitful exchanges with the authors, and the authors provided many further clarifications and new results. However, after the rebuttal phase, some major concerns (especially whether the proposed scaling law is a good fit empirically) still remains.

---

### Decision · Program_Chairs · 2025-01-22

Reject